# Are There Sex Differences in the Prevalence and Severity of Early-Stage Trauma-Related Stress in Mildly Impaired Autistic Children and Adolescents?

**Vicki Bitsika and Christopher Francis Sharpley \***

Brain-Behaviour Research Group, University of New England, Armidale, NSW 2350, Australia; vicki.bitsika@une.edu.au
\* Correspondence: csharpl3@une.edu.au; Tel.: +61-2-6773-2596

**Abstract:** There is some evidence that autistic children experience more traumatic events than non-autistic children, but little research attention has been given to sex differences on this issue. This study investigated the presence of sex-based differences in the occurrence and severity of trauma-related events and resultant stress in autistic youth, and tested the null hypothesis of no sex differences. A sample of 32 autistic males and 32 autistic females (6 yr to 18 yr), plus one of their parents, participated in a survey study of (a) the presence of a traumatic event and ongoing stress regarding that event, and (b) recurrent memories or dreams of that event. Although the autistic children rated their experience of trauma, plus their recurrent memories or dreams of that event, as more severe than their parents did, neither of these sources of information showed any significant sex differences in the total score or diagnostic frequency for trauma-related stress. There were no significant correlations between age, IQ, or autism severity and trauma-related stress scores for the autistic males or females. These results challenge the generalizability of the prevalence of sex differences in trauma-related stress that has been reported in the wider community, arguing that equal attention should be given to male and female autistic youth for this possible comorbidity.

**Keywords:** autism; sex differences; trauma; stress; age

## 1. Introduction

Autism Spectrum Disorder (ASD) [1] ('autism') is often association with comorbid psychiatric disorders [2], particularly anxiety [3,4], including some anxiety-related disorders that have been reclassified as Trauma- and Stressor-related Disorders (TSD) [1]. For example, one recent study reported an elevated rate of 32% of autistic adults suffering from the most severe form of TSD, i.e., Post-Traumatic Stress Disorder (PTSD), compared to just 4% of non-ASD adults sampled [5]. However, there is some uncertainty in the literature regarding the relative prevalence of PTSD in autistic youth [6–9]. In their recent review of this field, Ng-Cordell et al. ([10], p. 174) noted that PTSD was "under-investigated" in autistic youth, a finding that is congruent with the suggestion that there is a "pressing need" to further investigate the prevalence and correlates of TSD in autistic persons ([11], p. 290). There have been a number of recent comments regarding the possible contributors to TSD and PTSD in autistic persons [4,7,12], but some major issues remain unaddressed, and need resolution prior to the development of targeted treatment approaches.

First, age may be a major factor in TSD among autistic youth. Although there is a sex differential of about 2:1 for female: male prevalence of PTSD [13] in the wider population [1], this may be restricted to certain age groups. For example, a survey of 9966 (autism non-differentiated) children and adolescents in the UK found that there were no sex differences in PTSD prevalence for children aged 8 yr or 10 yr, but there were significant sex differences for adolescents aged 13 yr and 15 yr, suggesting a developmental profile for the female:male prevalence ratio of PTSD, although this was not clearly related to the onset of

puberty [14], nor focused upon autistic youth. Secondly, although most research on PTSD in ASD has focused upon adults [11], autistic children also often experience other TSDs, plus anxiety of various forms [15–18], perhaps due to having been maltreated or suffering some other trauma or major stressor [19]. Mehtar et al. [18] found that, of those autistic children and adolescents in their sample aged 6 yr to 18 yr who had experienced trauma, 25% of the females had developed full-blown PTSD but only 15% of the males fulfilled that diagnosis. However, no analyses of age subgroups were reported, so that the findings reported by Haag et al. [14] regarding an age effect in sex differences commencing at about age 13 yr in their non-autistic sample could not be tested. Thirdly, while PTSD is the major disorder in the TSD range, less-severe forms of TSD may be relevant to studies of the effects of trauma upon autistic youth because they may represent potential intervention points prior to full-blown PTSD. Thus, investigation of the prevalence of less-severe forms of TSD might provide further understanding of how widespread TSDs are among male and female autistic youth, with implications for the likely later development of PTSD. Fourthly, while the formal diagnosis of PTSD relies on (a) recurrent and disturbing memories or dreams of a major trauma or stressor, plus (b) changes in daily function (e.g., increased physiological arousal) [1], the definition of a traumatic event used in that nomenclature is 'exposure to actual or threatened death, serious injury, or sexual violence' ([1], p. 301). However, many autistic children may experience less severe forms of 'traumatic' events that give rise to recurrent disturbing memories or dreams of that event. For example, being bullied at school has been described by autistic boys as representing a major stressor that they cannot easily manage, and which recurs in their memories and/or dreams [20], but is not life-threatening. Alternately, Kerns, et al. ([6], p. 3475) argued that 'trauma' might be more broadly defined as 'a psychological injury' that may be minor, moderate, or severe, and which may lead to traumatic stress ('a persistent disturbance of mood, arousal and behavior following a traumatic event') or PTSD itself, a more severe outcome. Thus, 'traumatic event' is defined in this study according to Kerns et al.'s [6] description of any event that is associated with ongoing distress. Finally, because there is evidence that some ASD symptoms may be correlated with anxiety in autistic children [21], examination of the associations between the early stages of TSD (as defined above by the presence of a traumatic event, plus the recurrent disturbing memories or dreams of that event) and demographic factors such as IQ and ASD severity may also help describe the nature of any sex differences in TSD in autistic youth. This is important, because IQ has been shown to have an inverse correlation with autism severity [22], and there may be an interaction between IQ, autism severity, and TSD.

Because the characteristic symptoms of PTSD such as 'fear-based reexperiencing, emotional, and behavioral symptoms' ([1], p. 305) may take some time to appear, it may be valuable for the development of preventative treatment models to identify the early signs of TSD, termed "Early TSD" (ETSD) here. By describing the first stage of TSD (i.e., the presence of a traumatic event, the recurring disturbing memories or dreams of that event), the early warning signs of possible later TSD (and even PTSD) may be used to suggest appropriate preventative interventions to those autistic youth who may be at risk of consequent development of severe TSD.

Investigation of these issues could provide some clarity to the argument raised by Kerns et al. [23] regarding implementation of regular screening for PTSD in autistic youth. For example, if presence of the two early stages of TSD defined above was found, or found to differ between the sexes of respondents, then Kerns et al.'s [23] argument would be supported and extended to those early signs of TSD as well as formal assessment of PTSD itself, with possible implications regarding the sex of autistic youth. Therefore, the present study aimed to determine if there were any significant sex differences in the prevalence of two early signs of TSD in a sample of autistic youth of both sexes. Because of the lack of clear outcomes from the previous literature (as discussed above), the null hypothesis that there would be no significant sex-based differences in TSD was to be tested. Early signs of TSD were defined as presence of either or both (a) a history of having experienced a

major trauma or stressor, and (b) the symptom of having recurrent disturbing memories or dreams of that event. Data analysis was undertaken at the whole age range, and for subsamples aged 12 yr or younger versus those aged more than 12 yr to detect any age-related differences that were previously reported for PTSD in the wider population, i.e., by [14]. IQ and ASD severity were investigated for their association with these two signs of TSD because of their previous links to other anxiety-related disorders in autistic samples [24]. Although previous studies of autistic youth did not include examination of the effect of information source regarding the autistic child's ETSD symptoms, self-reports of autistic children's anxiety have previously been shown to agree more strongly with a biomarker of stress and anxiety than those given by their parents [25], and so autistic children's self-reports and parent-reports of the autistic youth's ETSD symptoms were also included to provide a comparison of source data in this sample. To reduce the possibility of age, IQ, and ASD severity confounding the comparison between males and females, the samples were matched on these three variables.

## 2. Materials and Methods

### 2.1. Participants

A total of 64 parents and their autistic children (32 males, 32 females) responded to a recruitment advertisement to parent support groups in the Gold Coast region of Queensland, Australia. Because some autistic children may exhibit sex dysphoria [26], the parents' identification of their child's sex at birth, plus the males' and females' own identification of their sex were used to confirm sex of these children. Although the call to recruitment did not specify it, all the parents of the autistic females, and 28 of the 32 parents of the autistic males were their mothers. All these autistic children had been previously diagnosed via clinical interviews with a paediatrician or psychiatrist, and a clinical psychologist. That diagnosis was confirmed by administration of the Autism Diagnostic and Observation Schedule-2nd edition (ADOS-2) [27] by a research-reliable assistant as part of the recruitment process for this study (all the autistic children received ADOS-2 Total scores of 7 or more). The Wechsler Abbreviated Scale for Intelligence (2nd edition) (WASI-II) [28] was administered by the same research-reliable assistant during recruitment, indicating that all the autistic children had Full Scale IQ scores of 70 or more. Because they were also attending mainstream schools, these autistic children were described as "mildly impaired" in the local context. None of the parents reported that their autistic children had been formally diagnosed with PTSD, anxiety, or depression, or were taking medication for these disorders. All these participants were recruited during a cross-sectional study of anxiety and depression among autistic children and adolescents who attended school [29] and the data reported here represent a subsample of the results of that study.

### 2.2. Instruments

Questionnaire. Parents completed a questionnaire about their child's age, diagnostic features, and history; the autistic girls' parents were also asked if their daughters had commenced menstrual cycling.

The Wechsler Abbreviated Scale of Intelligence (2nd edition) (WASI-II) [28] is a screening test of intelligence that possesses strong validity with the WISC-IV when used with high-functioning autistic youth [30]. It contains four subtests with average reliability coefficients of between 0.92 and 0.96, and these can be reduced to a Full Scale score useful for research screening purposes, referred to here as WASI-II FS.

The Autism Diagnostic Observation Schedule Second Edition (ADOS-2) [27] is recommended in several Best Practice Guidelines as an appropriate standardized diagnostic observation tool for ASD [31,32]. Results from this study were used to confirm a diagnosis of ASD via an ADOS-2 Overall Total SA + RRB score of 7 or more as recommended by the ADOS-2 authors [27]. This score was termed "ADOS-2 Total score" here as a measure of autism severity.

Child and Adolescent Symptom Inventory (4th ed.) (CASI-4R). The CASI-4R [33] is a screening test based upon DSM-IV-TR criteria for a range of disorders. It was normed in a variety of samples, including studies of 103 [34] and 67 autistic children [35]. Psychometric data are described in the CASI-4 Test Manual [33,36], and include test-retest reliability of $r = 0.67$ ($p < 0.001$) over a six-week period, internal consistency of 0.74, strong criterion-related validity with psychiatric diagnoses, internal construct validity, and discriminant validity [33]. Responses to the CASI-4R questionnaire items are 0 (never), 1 (sometimes), 2 (often), or 3 (very often) for items representing diagnostic criteria. The CASI-4R has parent and teacher scales, but previous research has indicated that data regarding autistic youth's anxiety can be collected from the youth themselves using the CASI-4R [37].

CASI-4R ETSD subscale. The CASI-4R has a subscale that screens for some PTSD symptoms. The authors of the CASI-4R state that it contains 'one screening item' for 'the most obvious features' of PTSD ([33], p. 18), namely "Has experienced an extremely upsetting event and continues to be bothered by it". The CASI-4R also includes the item "Has distressing memories or dreams about an extremely upsetting event", which also fits one of the diagnostic criteria for PTSD in the DSM-5-TR [1]. Although neither of these measures represent a formal diagnosis of PTSD, but rather an indication of the presence of ETSD ([33], p. 18), they do measure the two criteria for early stages of TSD as defined in the Introduction. The first of these two items was used to screen for the presence of the first criteria for ETSD, according to the procedures recommended by the authors of the CASI-4R (i.e., a score of "Sometimes" on the four-point response scale described above ([33], p. 18), and termed "CASI-4R ETSD Category", recorded as either present or absent. The second of these two CASI-4R items was also measured separately, and combined with the first item to produce a metric termed "CASI-4R ETSD score".

Source of ETSD data. The CASI-4R ETSD subscale was administered to the autistic children and to their parents, separately, by the research-reliable clinicians. Autistic children can self-evaluate their own emotions [38], including loneliness [39], anxiety [40], and depression [41].

### 2.3. Procedure

All participants were individually interviewed in their own homes by a research-reliable clinician, so as to reduce the likelihood of anxiety arising from being interviewed in an unfamiliar place (i.e., the authors' laboratory). Parents' and children's data were collected discretely from each other during the same interview period. The ADOS-2 and the WASI-II were administered to the autistic children by the research-reliable assistant during the three weeks immediately prior to the CASI-4R data collection, also in the participants' homes. Participants were informed that their responses would be kept confidential. Ethical approval for this study was obtained from the Bond University Human Research Ethics Committee (BUHREC) (No. 1516).

### 2.4. Data Analyses

Data were analysed with SPSS version 27 (IBM, Armonk, NY, USA). The CASI-4R ETSD data were tested for internal consistency and normality. Statistical significance and effect size were calculated. The principal statistical test used to detect associations between ETSD scores and the autistic participants' ages, WASI-II FS IQ, and ADOS-2 score was Pearson correlational analysis. MANOVA tested for differences in CASI-4R ETSD scores between male and female autistic youth, Chi-square was used to determine sex differences on the CASI-4R ETSD Category score, and *t*-tests were used to detect significant differences between parent- and self-reports on the two CASI-4R ETSD-related items. Effect sizes and statistical significance levels were referred to when determining the presence of significant and meaningful results. When necessary to reduce the likelihood of family-wise error, Bonferroni-adjusted *p* values were adopted to test for statistical significance. Because the major test of parent–child differences in estimates of ETSD was via *t*-test, an a priori power analysis was conducted, and indicated that, with a total sample of 64 participants, there

would be an 80% chance (power = 0.80) of detecting a significant effect size of 0.3 (medium strength) with $p < 0.05$.

## 3. Results

### 3.1. Sample Data

Internal consistency (Cronbach's alpha) for the CASI-4R ETSD 2-item subscale for parents was 0.801, and 0.738 for their autistic children's self-reports, which are unexpected for small scales of less than 10 items [42]. The inter-item correlation may be used instead of Cronbach's alpha [43] for small scales, and was 0.672 for the parents' data and 0.585 for the autistic children's data, both judged as acceptable [43]. The Normal Q-Q Plot for each CASI-4R ETSD 2-item subscale was a straight line and there were no outliers, and so no transformation of the raw data was undertaken. Mean and Standard Deviation data for age, WASI-II FS score, and the ADOS-2 for the autistic males and females are shown in Table 1, columns 1 to 3, indicating no significant sex differences on these variables. There were no significant correlations between age, WASI-II FS IQ, or ADOS-2 total score and either of the two sets of CASI-4R ETSD subscale scores (i.e., self- and parent-reports), nor for the presence/absence of ETSD, at the Bonferroni-adjusted $p$ value of 0.05/3 = 0.016.

**Table 1.** Mean (SD) for age, IQ, and ASD severity for 32 autistic males and 32 autistic females.

| Variable | Mean (SD) Males | Mean (SD) Females | $F$ | $p$ | $\mu^2$ |
|---|---|---|---|---|---|
| Age (yr) | 10.09 (3.82) | 10.31 (2.57) | 0.072 | 0.789 | 0.001 |
| WASI-II FS IQ [1] | 95.78 (13.67) | 99.97 (12.85) | 1.593 | 0.212 | 0.025 |
| ADOS-2 Total score [2] | 11.25 (2.92) | 12.40 (2.62) | 2.776 | 0.101 | 0.043 |

[1] Wechsler Abbreviated Intelligence Scale (2nd ed.) Full Scale score; [2] Autism Diagnostic Observation Schedule (2nd ed.).

### 3.2. Source Differences

Table 2 shows that the autistic youth rated their scores on the two CASI-4R ETSD items significantly more severely than their parents did.

**Table 2.** Self- versus parent-reports on two CASI-4R ETSD-related items.

| CASI-4R [1] Item | Self-Reports M (SD) | Parent-Reports M (SD) | $t$ | $p$ | Cohen's d |
|---|---|---|---|---|---|
| Has experienced an extremely upsetting event and continues to be bothered by it. | 1.52 (1.22) | 1.12 (0.98) | 3.279 | 0.002 | 0.361 |
| Has distressing memories or dreams about an extremely upsetting event | 1.20 (1.18) | 0.84 (0.67) | 2.605 | 0.011 | 0.375 |

[1] Child and Adolescent Symptom Inventory (4th ed.), Post-Traumatic Stress Disorder subscale.

### 3.3. Sex Differences

As shown in Table 3, there were no significant differences between autistic males and females for the CASI-4R ETSD score (composed of the two CASI-4R items described above in Methods), nor on the single CASI-4R item that the authors of the CASI-4R identified as a screening test of the presence of PTSD (referred to here as "CASI-4R ETSD category"), for either the autistic youth's self-scores or those given by their parents (although the ratings of severity of the CASI-4R ETSD subscale were greater for the children than their parents, reflecting the findings reported in Table 2). Of interest, 43.75% of the autistic males scored at least "Sometimes" on the parent-ratings of the CASI-4R screening item used to provide a

screening for PTSD, but Gadow and Spravkin [33] found that the percentage of 272 males in their normative sample who reached that screening criteria for PTSD was only 19.5%; for the autistic females in the current sample, 50% reached the 'Sometimes' criterion, much higher than the 12.5% reported by Gadow and Spravkin ([33], p. 63) for their normative sample of 279 females. However, those normative samples were composed of children aged between 6 yr and 12 yr, and so the next step in the data analysis provides a more focussed comparison with the norms for the CASI-4R.

**Table 3.** Mean (SD) scores for males and females for two measures of ETSD.

| Variable | Males | Females | *F* | *p* | $\mu^2$ |
|---|---|---|---|---|---|
| | Mean (SD) | Mean (SD) | | | |
| Self-scored CASI-4R ETSD score [1] | 2.28 (2.08) | 3.16 (1.14) | 2.745 | 0.103 | 0.042 |
| Parent-scored CASI-4R ETSD score | 1.78 (1.77) | 2.15 (1.66) | 0.759 | 0.387 | 0.012 |
| CASI-4R ETSD Category score [2] | Present | Absent | Present | Absent | Chi-square | *p* | *φ* |
| Self-scored | 14 | 18 | 18 | 14 | 0.563 | 0.453 | 0.125 |
| Parent-scored | 16 | 16 | 19 | 13 | 0.252 | 0.616 | 0.094 |

[1] Child and Adolescent Symptom Inventory (4th ed.), Post-Traumatic Stress Disorder subscale; [2] Single-item ETSD categorisation from CASI-4R.

When divided into the two age subgroups that were found to distinguish between the presence of sex differences in ETSD in the wider population [14], i.e., younger than 13 yr (Table 4a) versus 13 yr and older (Table 4b), there were also no significant sex differences in the ETSD Score or the prevalence of ETSD according to the CASI-4R ETSD score (composed of the two CASI-4R items described above in Methods), or the single-item CASI-4R ETSD Category results for either age subgroup. There were also no significant sex differences in age, WASI-II FS IQ, or ADOS Total Scores within either of these two age subgroups (all *p* values > 0.101). The cell sizes were small for the older autistic children, and therefore this analysis had less statistical power than would be desired. To address this limitation, these analyses were rerun with 1000 cases bootstrapping, but the results did not change. In reference to the normative data reported by Gadow and Spravkin [33] for their sample of 6 yr to 12 yr old females and males on the parent-rated CASI-4R single item (i.e., ETSD Category) criteria for possible ETSD, the current sample percentage for the males aged 6 yr to 12 yr who reached this criterion was 52.2%, and for the same age females it was 66.6%, both figures much higher than those reported by Gadow and Spravkin's [33] for males (19.5%) and females (12.5%).

**Table 4.** (**a**) Mean (SD) for ETSD scores for autistic males and autistic females aged less than 13 yr. (**b**) Mean (SD) for ETSD scores for autistic males and autistic females aged more than 12 yr.

| (a) | | | | | |
|---|---|---|---|---|---|
| Variable | Mean (SD) (yr) Males (*n* = 23) | Mean (SD) (yr) Females (*n* = 24) | *F* | *p* | $\mu^2$ |
| Self-scored CASI-4R ETSD score [1] | 2.43 (2.19) | 3.17 (2.20) | 1.305 | 0.259 | 0.028 |
| Parent-scored CASI-4R ETSD score [1] | 1.73 (1.68) | 2.25 (1.59) | 1.141 | 0.291 | 0.025 |
| CASI-4R Category [2] | Present | Absent | Present | Absent | Chi-square | *p* | *φ* |
| Self-scored | 12 | 11 | 13 | 11 | 0.189 | 0.778 | 0.063 |
| Parent-scored | 12 | 11 | 16 | 8 | 1.024 | 0.380 | 0.148 |

**Table 4.** *Cont.*

| (b) | | | | | | |
|---|---|---|---|---|---|---|
| Variable | Mean (SD) (yr) Males (*n* = 9) | | Mean (SD) (yr) Females (*n* = 8) | | $F$ | $p$ | $\mu^2$ |
| Self-scored CASI-4R ETSD score [1] | 1.88 (1.85) | | 3.12 (2.10) | | 1.680 | 0.214 | 0.101 |
| Parent-scored CASI-4R ETSD score [1] | 1.89 (1.72) | | 1.87 (1.59) | | 0.001 | 0.989 | 0.001 |
| CASI-4R Category [2] | Present | Absent | Present | Absent | Chi-square | $p$ | $\varphi$ |
| Self-scored | 3 | 6 | 5 | 3 | 1.446 | 0.317 | 0.292 |
| Parent-scored | 4 | 5 | 3 | 5 | 1.084 | 0.999 | 0.070 |

[1] Child and Adolescent Symptom Inventory (4th ed.), Post-Traumatic Stress Disorder subscale; [2] Single-item ETSD categorisation from CASI-4R.

Although Haag et al. [14] did not find that puberty was associated with PTSD prevalence, the possible association between autistic females' menarche status and their ETSD scores was investigated, but neither the single-item ETSD Category score nor the combined total of the two CASI-4R ETSD items (ETSD Score) were significantly associated with this measure of the autistic females' development (all $p > 0.328$).

## 4. Discussion

The first finding from this study was that this sample of autistic males and females demonstrated much higher rates of possible presence of ETSD than those reported by the authors of the CASI-4R in their normative data, based upon the single-item screening process described by Gadow and Spravkin [33]. Of note, despite there being a significant difference between the CASI-4R ETSD self-reports and parent-reports, both represented ETSD at a greater prevalence than the norming values reported in the CASI-4R Manual [33]. Of further relevance to this investigation, neither the self-reports nor the parent-reports demonstrated significant sex differences in the ETSD variables in this sample. Secondly, although the age-based comparison was necessarily restricted to the specific age group used in the norming sample for the CASI-4R (i.e., 6 yr to 12 yr), it is of interest that, unlike some previous data showing an increase in sex differences in PTSD prevalence among non-autistic adolescents [14], there was no such significant difference found in ETSD among the current subsample of 13- to 18-year-old autistic youth.

Although the reason for the departure of these data from the findings reported by Haag et al. [14] requires further investigation, possible hypothetical mechanisms might relate to a lack of difference in the frequency of traumatic events experienced by these autistic males and females respectively. That is, in Haag et al.'s [14] sample, females ages 13 yr and 15 yr reported having experienced more traumatic events than their similar aged male peers, and they also had higher PTSD prevalence. Based upon the much higher rate of ETSD found in the current study of autistic males and females than in previous reports of non-autistic males and females in general [1], and in non-autistic youth in particular via the CASI-4R [33], the difficulties that almost all autistic persons have with social interaction and comprehension at any age might prevent the decrease in PTSD symptoms that Haag et al. [14] observed in adolescent males (but not females).

This point also suggests the need for further investigation of the sources of ETSD in autistic youth. Social interaction and communication difficulties may represent (ongoing) traumatic events for autistic youth, but there are other sources of trauma that also require recognition. For example, the second major diagnostic criteria for ASD refers to repetitive and restrictive behaviour, and this may also bring social criticism upon autistic youth. Studies of bullying in autistic youth indicate that they experience bullying at greater rates than their non-autistic peers [44,45], occurring in 44% of autistic youth [46]. The kinds of bullying experiences that these young people report have been identified as being: the butt

of jokes, called names, excluded, hit, and ignored [20], which fit Kerns, et al.'s [6] definition of trauma. One of the major kinds of bullying received was "having rude things said about how they acted" ([20], p. 755), which may refer to the restrictive and repetitive behaviour characteristic of autism [1]. If that is the case, then it may be their autism itself which is associated with ETSD in autistic youth, although the lack of any significant correlations between ETSD scores and ADOS-2 Total scores does not support that hypothesis.

Haag et al. [14] cited several possible factors that may have contributed to the higher prevalence of PTSD in their sample of teenage females than males, such as the tendency for females to cope with stress via rumination [47], differences in biological vulnerabilities such as the HPA axis, neural circuits, and epigenetic factors [48–50], and differences in the amount of social support given to adolescent females than to males [51]. The present data that failed to show significant sex differences in ETSD prevalence suggest that these factors, which may contribute to sex differences in the wider population, may not apply within the autistic population. However, this is conjecture and requires further research.

The third research question posed for this study was in regard to the influence of IQ and ASD severity upon ETSD prevalence. The correlational analyses reported above did not reveal any such significant association between these demographic factors and ETSD prevalence or severity. By analysing ETSD data at the CASI-4R single-item (ETSD Category) level, as well as the sum of the two ETSD-related CASI-4R items (ETSD Score), the fourth research question (regarding the possible difference in outcomes from a dichotomous measure of ETSD presence/absence versus a linear score from two ETSD-related CASI-4R items) was able to be resolved as a nonsignificant source of difference. Finally, the significantly higher ETSD self-report scores given by the autistic children compared to those given by their parents contradicts previous findings for Generalised Anxiety Disorder, where parents rated their male [52] and female [53] children as more anxious than the autistic girls or boys did themselves, and suggests that the reclassification of ETSD away from the Anxiety Disorders [1] may reflect a pragmatic as well as theoretical difference in these two types of disorders, at least within this sample of autistic youth.

### 4.1. Implications for Clinical Practice

In terms of implications for clinical practice, these findings argue for regular screening for ETSD when assessing autistic children, supporting the recommendations made by Kerns, et al. [23]. This is of importance because one recent survey of 673 multidisciplinary autism spectrum disorder providers in the USA found that only 10% of those surveyed universally screened for the presence of PTSD when assessing youth with ASD [23]. If, as has been demonstrated here, ETSD occurs at a much higher prevalence for autistic youth than for the wider community, then clinician attention to this potential source of personal distress should be considered a high priority. In addition to the essential distress and symptoms associated with PTSD, such as irritability, exaggerated startle, difficulties concentrating, and sleep and communication problems [1], there are adverse outcomes from PTSD in terms of elevated risk of substance abuse [54], poorer employment outcomes [55], relationship problems [56], and increased risk of physical illness such as cardiovascular, metabolic, and musculoskeletal disorders [57]. These are major issues, and could only exacerbate the difficulties experienced by autistic persons. Early identification of PTSD at-risk autistic youth by assessing their ETSD could be a first step in the reduction of PTSD in these young persons.

### 4.2. Limitations and Future Research

There are some limitations upon the generalisability of these results. First, in terms of the dependent variable used here, the definition of ETSD is not specifically listed in the DSM-5-TR [1], but could fall under the categories of 'Other Specified Trauma- and Stressor-Related Disorder' or 'Unspecified Trauma- and Stressor-Related Disorder', which apply to the presentation of 'symptoms characteristic of trauma- and stressor-related disorders that cause clinically significant distress or impairment in social, occupational,

or other important areas of functioning but do not meet the full criteria for any of the disorders in the trauma- and stressor-related disorders diagnostic class' ([1], pp. 327–328). If either of these definitions were used, then the assessment process used in this study could be accepted as relevant, although caution led us to take the more conservative position of referring to 'early' TSD as a way of acknowledging the lack of more formal severe symptomatology that might occur (for example) if PTSD was the diagnosis used. A similar issue of definition may apply to the possibility of symptom overlap between ASD and PTSD that has been previously reported in several papers (e.g., [6,7,19]). Because PTSD was not the defined disorder examined here, this argument may have less of an influence on the validity of the dependent variable studied in this study. The measure used (the CASI-4R ETSD subscale) is a screening device only, and therefore caution must be applied to interpretation of the findings and their generalisability to other trauma- and stressor-related disorders, such as PTSD. Second, in terms of the sample studied, the recruitment of autistic youth who were 'mildly impaired' (IQ of 70+, attending mainstream schools) means that few conclusions can be drawn regarding autistic youth with lower IQ or greater autism impairment. Although volunteer participation is a reasonably common methodology for participant recruitment, it does not provide any insights into persons who chose not to participate. Third, in terms of research protocol, the ETSD data were collected at a single time point, and so generalisability over time is low, which also limits the ability to draw any causal inferences. Fourth, although justified by the power analysis, the sample size was relatively small, and further research with larger, more comprehensive samples would enhance the generalisability of these findings. Finally, data regarding family SES was not collected here, and this may be of value in future studies.

Despite these limitations upon the generalizability of these data to other samples of autistic youth and other trauma- and stressor-related disorders, the clear finding of a lack of significant and meaningful sex differences in scores on two types of measures of potential ETSD provides a new insight into the nature and widespread (i.e., for both sexes, in childhood and adolescence) experience of trauma and trauma- and stressor-related disorders in autistic youth. Additionally, the very high prevalence of possible ETSD here, compared to that reported for the general-population-based norming sample collected by the CASI-4R authors, argues strongly for clinical attention to this possible psychiatric comorbidity in autistic youth. Clearly, ETSD has a different prevalence profile in autistic youth, and further investigation of the nature of that difference, apart from elevated prevalence, is a research priority.

**Author Contributions:** Conceptualization, V.B. and C.F.S.; methodology, V.B.; software, C.F.S.; validation, V.B. and C.F.S.; formal analysis, C.F.S.; investigation, V.B.; resources, V.B.; data curation, V.B.; writing—original draft preparation, C.F.S.; writing—review and editing, C.F.S. and V.B.; visualization, V.B.; supervision, V.B.; project administration, V.B.; funding acquisition, V.B. All authors have read and agreed to the published version of the manuscript.

**Funding:** This research received no external funding.

**Institutional Review Board Statement:** The study was conducted in accordance with the Declaration of Helsinki, and approved by the Institutional Review Board (or Ethics Committee) of Bond University Human Research Ethics Committee (protocol code 1516, 15 September 2013).

**Informed Consent Statement:** Informed consent was obtained from all subjects involved in the study. Written informed consent has been obtained from all participants to publish this paper.

**Data Availability Statement:** Data are available from the first author on reasonable request.

**Conflicts of Interest:** The authors declare no conflict of interest.

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
