# Peer review of "Are There Sex Differences in the Prevalence and Severity of Early-Stage Trauma-Related Stress in Mildly Impaired Autistic Children and Adolescents?"

_traumacare, doi:10.3390/traumacare3040023_

Round 1
Reviewer 1 Report
Comments and Suggestions for Authors
Brief Summary
Title: Are there sex differences in the prevalence and severity of early-stage trauma-related stress in mildly impaired autistic children and adolescence?
Summary: The authors surveyed 32 female and 32 male mildly autistic children, ages 6-18, along with one parent examining the presence of a traumatic event, ongoing stressors about said event, and recurrent symptoms of stress associated with that event, e.g., recurrent memories, dreams of the event. They believed based on past literature that their sample would show more female children showing symptoms of trauma as compared to male children. They also examined whether severity of autistic symptoms would correlate strongly with trauma-related scores. The authors found that autistic children were more likely to rate their early stages of trauma as more severe as compared to their parents. This result is interesting, and I believe new to the literature. The authors also did a nice job talking about the issues within the DSM for diagnostic criteria for Acute Stress Disorder and Post Traumatic Stress Disorder. Their particular sample had symptoms of stress despite not meeting the criteria for PTSD. They did not show sex differences in trauma for their sample.
While I recognize the difficulty in finding a large diverse sample of autistic children and their parents who are willing to participate in the study, this represented a small sample, making it difficult to really answer the question about level of severity and trauma. There wasn’t enough diversity in the level of severity to really answer this question.
Another issue that wasn’t considered in the manuscript had to do with the demographics of the families studied. More needs to be reported on the background of the families, e.g., demographics. From reading the manuscript, I am wondering if the families were higher SES, more supportive, etc. Again, the sample does not seem as diverse as would be nice to examine the hypotheses in earnest. Further, the sample size was small.
In the discussion, I think they make a good point about clinicial significance - that while generalizability of this study is low, asking about ETSDs and PTSD as well as other psychiatric co-morbidities is warranted.
Author Response
We acknowledge the restricted sample size, and have now added a power analysis to lines 203-207, justifying the sample we had. However, the sample was still relatively small, so we have added the following statement to the end of the Limitations, lines 393-395: "Fourthly, although justified by the power analysis, the sample size was relatively small, and further research with larger, more comprehensive samples would enhance the generalisability of these findings."
We have also added the following statement to the Limitations to reflect our lack of data on family SES, lines 395-396: "Finally, data regarding family SES was not collected here, and this may be of value in future studies."
Reviewer 2 Report
Comments and Suggestions for Authors
Thank you very much for this well-written paper. the method section was clear and informative, the rational provided in the introduction too.
I only have a few minor issues
in table 3 one sees that parent-scored and self-scored presence of a trauma is well-aligned, i.e. the difference is in the severity rating, not the knowledge of a traumatic event. That, IMO, illustrates that it is self-perceived vs others that differs between the children and their parents. (see also table 4b where presence rating is higher by parents than autistic female adolescents)
line 185: you write "parents' and their son's - should it not be parents and their child?
very minor: there are a lot of space issues
Author Response
We thank the reviewer for pointing out that the major difference between parents and children in Table 3 was for ratings of ETSD severity, not for the presence/absence. We have now made an additional point about that, although that aspect of the data analysis was reported in Table 2. See lines 236-238: "although the ratings of severity of the CASI-4R ETSD subscale were greater for the children than their parents, reflecting the findings reported in Table 2)."
We have corrected the statement at line 185 to read: "Parents and children's data".
We have corrected space errors in the ms.
Round 2
Reviewer 1 Report
Comments and Suggestions for Authors
The authors have addressed my concerns in the manuscript.
Author Response
Thank you.
Reviewer 2 Report
Comments and Suggestions for Authors
Thank you for the edits.
Author Response
Thank you.